# Use of social media for cancer prevention and early diagnosis: scoping review protocol

Aradhna Kaushal ![ORCID] ,[1] Angelos P Kassianos,[2] Jessica Sheringham,[2] Jo Waller,[1] Christian von Wagner ![ORCID] [1]

¹Department of Behavioural Science and Health, University College London, London, UK
²Department of Applied Health Research, University College London, London, UK

**Correspondence to**
Dr Aradhna Kaushal;
aradhna.kaushal.14@ucl.ac.uk

## ABSTRACT

**Introduction** Social media platforms offer unique opportunities for health promotion messages focusing on cancer prevention and early diagnosis. However, there has been very little synthesis of the evaluation of such campaigns, limiting the ability to apply learning to the design of future social media campaigns. We aimed to provide a broad overview of the current research base on social media interventions for cancer prevention and early diagnosis, to identify knowledge gaps and to inform policy, practice and future research questions.

**Methods** We will use scoping review methodology to explore the available evidence on social media interventions for cancer prevention and early diagnosis, with a focus on methodological approaches. Quantitative and qualitative studies and reports will be identified through searching several research databases, through internet searching for grey literature and by screening the citations of studies included in the review. All identified studies will undergo independent title and abstract screening and full-text screening against inclusion and exclusion criteria. We plan to chart the data from included studies to record the characteristics of the social media interventions, resources, activities, outputs, outcomes and impact. Charted data will be collated and summarised using a narrative synthesis. The interpretation and implications of the findings will be enhanced by consultation with relevant stakeholders such as public health organisations, cancer charities, and patient and public involvement groups when preliminary results are available.

**Ethics and dissemination** Ethical approval is not required for this scoping review. The results will be used to identify research questions for future systematic reviews and to inform the development of future social media interventions. We will disseminate findings in peer-reviewed journals and at relevant conferences.

## Strengths and limitations of this study

► This scoping review will allow an exploration of the research evidence on the use of social media platforms for cancer prevention and early diagnosis. This will identify gaps in the literature and areas for future research.

► Due to the fast-moving nature of this research area, this scoping review will provide a timely synthesis of current research evidence.

► Obtaining insights into the mechanisms by which social media campaigns achieve impact will help inform the design of future interventions.

► Consultations with stakeholders will improve the impact and relevance of the scoping review.

► There is no formal assessment of risk of bias in scoping reviews, limiting our understanding of the quality of the research included.

on Cancer Campaigns in England, which aim to improve early diagnosis of cancer by raising awareness of cancer symptoms and encouraging prompt help-seeking from a general practitioner.[3] These campaigns have shown some modest increases on area-level knowledge of cancer symptoms and on earlier stage of lung cancer diagnosis.[4 5] However, there are concerns that the effects of these mass media campaigns are short-lived and prompt the 'worried well' to seek help.[6 7]

Social media, also known as Web 2.0 or the participatory or social web, is characterised by a movement from information storage towards user-generated content.[8] These platforms offer a novel and unique opportunity to improve cancer outcomes by encouraging help-seeking for common symptoms and by increasing awareness and uptake of cancer screening.[9–13] Neiger *et al* propose five main purposes of social media in public health and health promotion: (1) to communicate with the audience to gain insights; (2) to establish and promote a brand; (3) to disseminate critical information; (4) to expand reach to include broader, more diverse audiences; and

## INTRODUCTION

Despite recent improvements in cancer survival, marked differences remain between countries with similar healthcare systems.[1] These differences may be explained by low public awareness, barriers to help-seeking and screening, and negative beliefs about cancer.[2] This has prompted several national mass media campaigns, such as the Be Clear

(5) to foster public engagement and partnership with consumers.[14]

Social media use is becoming ubiquitous with recent data from the Office for National Statistics showing that the proportion of people using the internet for social networking has increased from 45% in 2011 to 65% in 2018.[15] This shift in the way people use the internet makes investigating methods for engaging the public with health messages via social media a high priority.[9] Public health campaigns delivered via social media platforms can influence behaviour through the same mechanisms as mass media campaigns by directly targeting cognitive or emotional responses, generating discussions within communities and changing social norms.[16] In addition, social media provides a platform for health messages to be disseminated rapidly and communicated through online networks and interactive communication between individuals.[17–21]

There are several advantages for using social media platforms for health promotion messages. A particular advantage is the ability to target messages towards specific geographical regions or hard-to-reach demographic groups, such as those who are known to have poor knowledge of symptoms or more barriers to accessing cancer screening.[10 22] Social media can therefore facilitate a more targeted approach to addressing health inequalities than is possible using more traditional mass media campaigns.[23 24] Social media platforms also have the potential for instantaneous and real-time communication between patients and healthcare providers while also providing features to track and monitor communication content saved on platforms.[25] Finally, moving public health campaign messages online may be a more pragmatic approach to reach teenagers and young adults for whom social media is an integral part of daily life.[26]

As well as these advantages, social media also present a certain number of challenges when it comes to promoting public health messaging. Very little is known about the unintended effects of social media campaigns, and disengagement or avoidance of communication messages is not routinely measured or analysed. There is also the possibility of misinformation spreading quickly via social media platforms as recently shown by antivaccination propaganda.[27 28]

Despite the widespread use of social media interventions for public health campaigns, little is known about the scientific evidence of these interventions. Understanding the components of successful social media interventions and how these might interact will help to inform the development of new interventions. Social media networks are a relatively recent phenomenon which has attracted a high-level research interest, but very little synthesis has occurred. This may be partially due to the challenges of evaluating outcomes: most evaluation methods use metrics such as number of views or shares, which may not indicate a genuine increase in knowledge or change in behaviour. Many of the campaigns are not formally evaluated, and reports on social media campaigns are often not publicly available, which further contributes to the challenge of synthesising the research evidence.

We propose a scoping review to explore the use of social media interventions to raise cancer awareness, promote early diagnosis of cancer and increase uptake of cancer screening. A scoping review can be defined as an 'exploratory project that systematically maps the literature available on a topic, identifying key concepts, theories, sources of evidence and gaps in the research'.[29] We decided a scoping review methodology best addressed our overarching aim, which is to provide a broad overview of the current research base on social media interventions, to identify knowledge gaps, and to inform policy, practice and future research questions.[30] We will also address the methodological opportunities and challenges in evaluating the effectiveness of social media interventions, for example, how to maximise what we can learn from the volume of available data collected by such technologies. We will develop a theoretical model, which proposes how social media interventions could promote the prevention and early diagnosis of cancer. To do this, we will conduct a timely synthesis of quantitative and qualitative studies to identify methodological approaches to evaluation, focusing on outcomes measured, and to understand how and why such interventions have achieved their effects.

## METHODS

We have used the scoping review methodology outlined by Arksey and O'Malley and expanded on by Levac *et al* and Peters *et al* to inform the development of this scoping review protocol.[31–33] The framework comprises six stages: (1) identifying the research question; (2) identifying relevant studies; (3) study selection; (4) charting the data; (5) collating, summarising and reporting the result; and (6) consultation. We also used the recently published Preferred Reporting Items for Systematic Reviews and Meta-Analyses Extension for Scoping Reviews to guide the development of this protocol.[34] We expect to complete stages 1–3 by October 2019, stage 4 by February 2020 and stages 5 and 6 by May 2020.

### Stage 1: identifying the research question
The proposed research question is: What is known from the existing literature about social media interventions, aimed at cancer prevention and early diagnosis of cancer? Specifically, we seek to answer the following questions with respect to these social media interventions:
1. What methodological approaches have been used to evaluate social media interventions?
2. What are the outcomes used to measure the impact of social media interventions?
3. What are the mechanisms through which social media interventions operate?

### Stage 2: identifying relevant studies
Relevant studies will be identified by searching research databases Medical Literature Analysis and Retrieval

## Box 1    Example search strategy

Medical Literature Analysis and Retrieval System Online (OVID) and Epub Ahead of Print, In-Process and Other Non-Indexed Citations and Daily (1946–6 June 2019)
1. Social media/ (5759)
2. online social networking/ (30)
3. 'web 2.0'.mp. (624)
4. facebook.mp. (2868)
5. twitter.mp. (2415)
6. instagram.mp. (282)
7. pinterest.mp. (49)
8. snapchat.mp. (41)
9. youtube.mp. (1448)
10. 1 or two or three or four or five or six or seven or eight or 9 (9959)
11. exp *Health Knowledge, Attitudes, Practice/ (56098)
12. exp *Neoplasms/ (2761991)
13. 11 and 12 (5060)
14. early diagnosis/ or 'early detection of cancer'/ (44870)
15. ((cancer* or tumo?r* or malignan* or neoplasm*) adj3 (awareness* or appraisal* or knowledge or informed or recogn*)).mp. (20631)
16. ((cancer* or tumo?r * or malignan* or neoplasm*) adj3 (symptom* or 'red flag')).mp. (13786)
17. ((cancer* or tumo?r * or malignan* or neoplasm*) adj3 screen*).mp. (46196)
18. ((cancer* or tumo?r * or malignan* or neoplasm*) adj3 (diagnos?s* or stag?* or detect*)).mp. (384121)
19. 13 or 14 or 15 or 16 or 17 or 18 (460408)
20. 10 and 19 (94)
21. limit 20 to English language (94)

System Online, PsycINFO, Scopus, Web of Science and the Cumulative Index to Nursing and Allied Health Literature by using predefined search strategies, including a combination of subject headings (eg, Medical Subject Headings and Emtree) and keyword searches, truncation, wild cards and proximity functions. The research team and an experienced research librarian developed the search strategy (box 1). The search will be limited to studies published since 2004 as this is considered to be the advent of widespread social media use. We will only include studies which are published in English as we do not have the resources to translate studies. After searching the databases, duplicates will be removed and imported into a spreadsheet for screening.

Additional studies will be identified by conducting internet searches for relevant published material and by hand-searching reference lists of included studies for other eligible studies. We will also contact organisations and charities which run social media interventions in early diagnosis of cancer, cancer awareness and cancer screening to request any relevant reports or papers.

### Stage 3: study selection

All identified studies will undergo two stages of screening: abstract screening and full-text screening. Two researchers will independently review the abstracts of all studies against the inclusion criteria. Prior to the screening process, the inclusion criteria will be piloted on a sample of abstracts to ensure they are appropriate for the type of studies screened. This process will be repeated for the full-text screening stage. Researchers will communicate throughout the screening process to discuss any challenges arising in study selection and clarify the inclusion and exclusion criteria if necessary. Any disagreements will be decided on by discussion or resolved by a third researcher. Inter-rater reliability for both stages of screening will be calculated using Cohen's kappa statistic.[35]

To be included in the review, studies should take place within the context of social media platforms, limited to Facebook, Twitter, Instagram, YouTube, Pinterest or Snapchat. Studies which report findings from mixed media campaigns (eg, television and radio) will be included if social media is one of the reported modes of communication. The primary message of the intervention should focus on potential symptoms of cancer, promotion of help-seeking for potential cancer symptoms, and awareness of or promotion of cancer screening. We will exclude review articles and studies in which the participants are patients with cancer or cancer survivors or if the intervention is targeted at health professionals. We will also exclude studies that evaluate static internet pages, such as blog posts, which do not have a social media element. We will include both qualitative and quantitative studies which report data relevant to the data charting fields described in stage 4 or if there is any information reported which is deemed relevant to the research questions.

We have decided to include both qualitative and quantitative studies in this scoping review as both will provide different insights; while quantitative studies will shed some light on the metrics used to evaluate social media campaigns on health outcomes, we anticipate that qualitative studies will provide information about the mechanisms through which these campaigns operate.

### Stage 4: charting the data

We will chart the data using a database entry form in Microsoft Access to collect key information on study characteristics, methodological approaches and outcome measures. We will also chart data on the mechanisms through which social media interventions work by basing the data entry fields on a logic model, a model commonly used to summarise the process of how an intervention leads to its desired outcomes.[36] For each study, this includes charting data on the resources required, activities undertaken, outputs, outcomes and impact (table 1).

We will develop guidance documentation, which elaborates on each field of the data charting process in order to improve inter-rater agreement. We will pilot this document alongside the data charting form to ensure the tools are comprehensive and appropriate for the included studies. Data charting will be completed by one researcher and verified by a second researcher with regular meetings with the study team throughout the process to discuss any iterative changes to the data charting fields.

**Table 1** Data charting domains and description

| Data charting domain | Description |
| --- | --- |
| Study information | Reference information<br>Country<br>Aim<br>Study type (observational, experimental and qualitative)<br>Social media platform<br>Target audience details (generation/sociodemographic variables)<br>Comparison group details<br>Cancer type |
| Resources | Staff<br>Planning time<br>Networks<br>Designers/marketers<br>Content development |
| Activities | Campaign launch<br>Set-up and manage community groups/social media feeds and accounts<br>Building partnerships and relationships<br>Promotion activities<br>Public relations |
| Outputs | Exposure<br>Reach<br>Insights<br>Engagement (low/medium/high) measured using key performance indicators as defined by Neiger et al[14] |
| Outcomes | Changes in awareness<br>Changes in intention<br>Changes in behaviour<br>Changes in health outcomes |
| Impact | Earlier diagnosis of cancer<br>Cancer prevention<br>Cancer mortality |

## Stage 5: collating, summarising and reporting the results

We will present a descriptive summary of each of the included studies to describe the characteristics of the study. Traditional methods of knowledge synthesis used in systematic reviews are likely to be inappropriate for analysing complex evidence, which is heterogeneous in its methodology and reporting.[37] There are several methods reported in the literature for synthesising such data; however, many of these methods are poorly defined and many have overlapping concepts, resulting in a lack of consensus on the most suitable method.[38]

We propose conducting a narrative synthesis using guidance produced by the Economic and Social Research Council Methods Programme.[39–41] There are four elements to the process of a narrative synthesis: (1) developing a theory of how, why and for whom the intervention works; (2) developing a preliminary synthesis of findings; (3) exploring relationships in the data; and (4) assessing the robustness of the synthesis. We will synthesise quantitative and qualitative data using a convergent synthesis design where both types of data are analysed concurrently.[42 43] Depending on the type of data charted from these studies, we will consider whether quantitative and qualitative data are analysed together using the same methods (data-based convergent design) or separately using different methods (results-based convergent design).

The exact format for the presentation of the results will depend on the nature of findings. We will explore the suitability of different methods for reporting the results such as narrative summaries, tables, figures and conceptual maps.[33] These results may be used to inform the development of guidelines for content development for social media interventions on cancer prevention and early diagnosis, in addition to a summary of different methodological approaches and an evaluation framework for such interventions.

## Stage 6: consultation

We will consult with stakeholders throughout the scoping review process (1) to identify relevant literature to include in the review and (2) to contribute towards the interpretation and policy implications of the results. We have identified several stakeholders from public health bodies, cancer charities, patient and public involvement groups, social media experts and researchers in cancer awareness with additional stakeholders to be identified via professional networks. When preliminary results are available, we will conduct focus groups and/or surveys with stakeholders to gain an understanding of different perspectives and interpretations of the findings. The results of these consultations will directly contribute to the interpretation of findings and conclusions in the final report.

## Patient and public involvement

This project was approved by the executive board of the National Institute for Health Research Policy Research Unit in Cancer Awareness, Screening and Early Diagnosis, which includes a patient and public involvement representative. We plan to include patient and public involvement groups in the interpretation and dissemination of the scoping review results.

## Ethics and dissemination

Ethical approval is not required for this scoping review. We plan to disseminate the results of this review in a peer-reviewed journal and at relevant conferences. The results of this review will help to identify gaps in the research literature and to optimise and inform the design of future social media interventions for cancer prevention and early diagnosis.

**Contributors** AK, JS, JW and CvW conceptualised the project. AK, APK JS, JW, and CvW contributed to developing the research question, refining the study methodology and contributed meaningfully to the drafting and writing of the final protocol.

**Funding**  This report presents independent research commissioned and funded by the National Institute for Health Research (NIHR) Policy Research Programme, conducted through the Policy Research Unit in Cancer Awareness, Screening and Early Diagnosis (PR-PRU-1217–21601). JS is supported by the National NIHR Collaboration for Leadership in Applied Health Research and Care North Thames at Barts Health NHS Trust.

**Disclaimer**  The views expressed are those of the authors and not necessarily those of the National Institute for Health Research, the Department of Health and Social Care or its arm's length bodies or other government departments.

**Competing interests**  None declared.

**Patient consent for publication**  Not required.

**Provenance and peer review**  Not commissioned; externally peer reviewed.

**ORCID iDs**
Aradhna Kaushal http://orcid.org/0000-0002-3815-0624
Christian von Wagner http://orcid.org/0000-0002-7971-0691

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
