## [Reviewer comments · BMJ Open]

ARTICLE DETAILS

TITLE (PROVISIONAL)	Use of Social Media for Cancer Prevention and Early Diagnosis: A Scoping Review Protocol
AUTHORS	Kaushal, Aradhna; Kassianos, Angelos P; Sheringham, Jessica; Waller, Jo; von Wagner, Christian

VERSION 1 – REVIEW

REVIEWER	Ravi Mehrotra ICMR_ICRC, India
REVIEW RETURNED	06-Nov-2019

GENERAL COMMENTS	Kindly include LinkedIn as one of the social media platforms, if feasible
---

REVIEWER	Desiree Schliemann Queen's University Belfast, United Kingdom
REVIEW RETURNED	09-Dec-2019

GENERAL COMMENTS	This is a well written and thought through protocol and I look forward to reading the review. I like the approach the authors take and Table 1 looks like the results will be very informative for future social media interventions. There are some few minor comments that I would encourage the authors to address. 1. The introduction starts with 'cancer survival in the UK' which leaves the reader to believe that this review is only looking at studies that have been conducted in the UK. Is this the case? If so, this needs to be mentioned in the methods, if not, I would encourage the authors to change the start of the introduction.2. Please make sure to spell out all abbreviations (GP, PPI).3. The references included in the review are relevant, but there are more recently published reviews on social media interventions that might be worth mentioning in the introduction.4. In the abstract intro - change past tense 'aimed' to 'aim'5. BMJ Open requires for the dates of the study to be included, it would be useful to include the timeframe for the review.6. Page 9 line 4 add - ... reliability for both stages of screening will 'be' calculated7. Do the authors include social media interventions targeted at health professionals? (if not please include in exclusion criteria).8. Do the authors consider 'mixed media' studies for inclusion. Many mass media campaigns may utilise different types of media, eg TV and social media. If that's the case, 'mass media' campaigns should be included in the search as well as added to the search terms. Please clarify in the methods section.
--

	9. Page 10 line 59 - grammar - please rephrase: 'will be depend on'
--	---

VERSION 1 – AUTHOR RESPONSE

Reviewer: 1

Reviewer Name: Ravi Mehrotra

Institution and Country: ICMR_ICRC, India Competing Interests: None

1. Kindly include LinkedIn as one of the social media platforms, If feasible.

Thank you for your suggestion. We did not consider LinkedIn as a potential platform for disseminating a public health message as it primarily used for professional networking and we are excluding interventions targeted at health professionals.

We re-ran the search, this time including LinkedIn as a search term and did not find any additional papers.

Reviewer: 2

Reviewer Name: Desiree Schliemann

Institution and Country: Queen's University Belfast, United Kingdom Competing Interests: None declared

This is a well written and thought through protocol and I look forward to reading the review. I like the approach the authors take and Table 1 looks like the results will be very informative for future social media interventions. There are some few minor comments that I would encourage the authors to address.

1. The introduction starts with 'cancer survival in the UK' which leaves the reader to believe that this review is only looking at studies that have been conducted in the UK. Is this the case? If so, this needs to be mentioned in the methods, if not, I would encourage the authors to change the start of the introduction.

We have changed the start of the introduction as suggested.

“Despite recent improvements in cancer survival, marked differences remain between countries with similar healthcare systems.”

2. Please make sure to spell out all abbreviations (GP, PPI).

We have checked the manuscript for abbreviations which have now all been spelled out.

3. The references included in the review are relevant, but there are more recently published reviews on social media interventions that might be worth mentioning in the introduction.

We have included references to more recent reviews in the introduction:

- Schliemann D, Su T, Paramasivam D, Donnelly M. Effectiveness of Mass and Small Media Campaigns to Improve Cancer Awareness and Screening Rates in Asia: A Systematic Review. American Society of Clinical Oncology; 2018.
- Welch V, Petkovic J, Simeon R et al. Protocol: Interactive social media interventions for health

behaviour change, health outcomes, and health equity in the adult population. 2018.

• Shi J, Poorisat T, Salmon CT. The use of social networking sites (SNSs) in health communication campaigns: review and recommendations. *Health communication*. 2018;33(1):49-56.

4. In the abstract intro - change past tense 'aimed' to 'aim'

This had been changed.

5. BMJ Open requires for the dates of the study to be included, it would be useful to include the timeframe for the review.

Timelines have now been included in the manuscript.

"We expect to complete stages 1-3 by October 2019, stage 4 by February 2020 and stages 5-6 by May 2020."

6. Page 9 line 4 add - ... reliability for both stages of screening will 'be' calculated

This has now been changed.

7. Do the authors include social media interventions targeted at health professionals? (if not please include in exclusion criteria).

We have excluded interventions targeted at health professionals. This exclusion criteria have been clarified.

"We will exclude review articles, and studies in which the participants are cancer patients or cancer survivors or if the intervention is targeted at health professionals."

7. Do the authors consider 'mixed media' studies for inclusion. Many mass media campaigns may utilise different types of media, eg TV and social media. If that's the case, 'mass media' campaigns should be included in the search as well as added to the search terms. Please clarify in the methods section.

We intend to include mixed media studies if social media is one of the modes of communication. This has been clarified in the methods:

"Studies which report findings from mixed media campaigns (e.g. television and radio) will be included if social media is one of the reported modes of communication."

Thank you for your suggestion to include 'mixed/mass media campaigns' in the search strategy. We have decided not to include these terms as it would greatly increase the number of articles for screening. Results from preliminary screening shows that the majority of eligible papers are from mass/mixed media campaigns and these types of campaigns will be well represented in the review.

9. Page 10 line 59 - grammar - please rephrase: 'will be depend on'

This has been corrected to 'will depend on'.